# Mathematical Modelling of Ultrasound-Assisted Extraction Kinetics of Bioactive Compounds from Artichoke By-Products

**DOI:** 10.3390/foods10050931

**Published:** 2021-04-23

**Authors:** Cristina Reche, Carmen Rosselló, Mónica M. Umaña, Valeria Eim, Susana Simal

**Affiliations:** Department of Chemistry, University of the Balearic Islands, Ctra. Valldemossa km 7.5, 07122 Palma de Mallorca, Spain; cristina.reche@uib.es (C.R.); carmen.rossello@uib.es (C.R.); monica.umana@uib.es (M.M.U.); valeria.eim@uib.es (V.E.)

**Keywords:** artichoke, by-products, ultrasound-assisted extraction, phenolic compounds, chlorogenic acid, diffusion model

## Abstract

Valorization of an artichoke by-product, rich in bioactive compounds, by ultrasound-assisted extraction, is proposed. The extraction yield curves of total phenolic content (TPC) and chlorogenic acid content (CAC) in 20% ethanol (*v/v*) with agitation (100 rpm) and ultrasound (200 and 335 W/L) were determined at 25, 40, and 60 °C. A mathematical model considering simultaneous diffusion and convection is proposed to simulate the extraction curves and to quantify both temperature and ultrasound power density effects in terms of the model parameters variation. The effective diffusion coefficient exhibited temperature dependence (72% increase for TPC from 25 °C to 60 °C), whereas the external mass transfer coefficient and the equilibrium extraction yield depended on both temperature (72% and 90% increases for TPC from 25 to 60 °C) and ultrasound power density (26 and 51% increases for TPC from 0 (agitation) to 335 W/L). The model allowed the accurate curves simulation, the average mean relative error being 5.3 ± 2.6%. Thus, the need of considering two resistances in series to satisfactorily simulate the extraction yield curves could be related to the diffusion of the bioactive compound from inside the vegetable cells toward the intercellular volume and from there, to the liquid phase.

## 1. Introduction

Large quantities of by-products are produced through the processing of plant-based foods whose re-introduction into a circular sustainable economy is important for reducing environmental problems generated by agro-industrial processes [1,2]. Moreover, in recent decades there has been considerable interest in the food industry in finding new sources of bioactive compounds with health and nutritional benefits. One of these sources is vegetable by-products because they are usually rich in antioxidants and food fiber [3].

Artichokes (*Cynara scolymus* L.) are widely grown and consumed in the Mediterranean region, and as such are considered a fundamental food of the Mediterranean diet [4]. During the industrial processing of artichoke, about 80–85% of the total plant biomass is rejected, being the heart of the artichoke the edible part [5]. This waste consists mainly of the artichoke bracts, stems, and leaves. Artichokes are a rich source of phenolic compounds, mainly chlorogenic acid, cynarine, caffeoylquinic acid, and flavonoids [6,7]. These compounds are responsible for the high antioxidant activity (16.76–180 mg of Trolox equivalents/g dry matter, dm) and other properties such as hepatoprotective, choleretics, anticarcinogenic, anti-HIV, diuretic, antifungal, and antibacterial properties [8,9]. Chlorogenic acid is a major artichoke phenolic compound, displaying a strong scavenging activity against reactive oxygen species and free radicals, and showing protective activity against oxidative damage [10].

For the recovery of bioactive compounds retained in the vegetable matrix, solid-liquid extraction is one of the most commonly used unit operations in the food industry [11]. One of the important variables that can be optimized to minimize the energy cost of the process is the temperature [12]. This variable affects not only the equilibrium and the mass transfer rates during the extraction but also, the yield in thermally labile compounds, so, a sufficiently high temperature can contribute to their degradation [13]. Standard extraction techniques usually use large amounts of organic solvents and imply high operating costs and long extraction times [14]. Thus, intensification of the extraction process is one of the areas with the greatest potential for development in the food industry. Among the techniques proposed to do this, power ultrasound is one of the most promising, allowing higher yields in shorter times and with lower energy costs [15].

The intensification of the extraction process by power ultrasound is related to several phenomena occurring simultaneously. One of the most important and studied is the cavitation phenomenon, which consists of the formation, growth and implosion of gas nano-microbubbles in the liquid, causing the molecules to oscillate around their equilibrium position, producing intermolecular distances to be continuously modified following alternate cycles of compression and decompression [16]. The implosion of cavitation bubbles leads to macro-turbulences and micromixing in the liquid media and microjets on the solid surface, which provokes surface peeling, erosion, and particle breakdown [17]. Furthermore, sonoporation, the cell walls’ permeabilization, and the sonocapillary effect, both due to ultrasound, can increase the release of cellular content into the extractive medium.

Overall, the effects of the ultrasound on the solid/liquid system can alter the cell walls microstructure [18]. Vallespir et al. [19] observed the formation of micro-channels in mushroom cell walls after ultrasound-assisted convective drying, which were wider when the temperature increased. Llavata et al. [16] described in a review that according to a number of different authors, ultrasound treatment causes the weakening of the cellular structure and the creation of microcracks, which facilitate the mass transfer.

Mathematical modelling allows us to estimate, control and predict the behavior of a process under different extraction conditions; at the same time, it can contribute to the optimization of the process [20].

When two phases with different compositions, as in an extraction process, come into contact, the diffusing substance leaves its place in an area of high concentration and moves to an area of low concentration. To understand the relationship between the parameters involved in ultrasound-assisted extraction and the efficiency of the antioxidant compounds’ recovery from natural sources, different modeling strategies have been proposed, most of them, by using empirical or kinetic models [21]. One of the proposed approaches considers that the extraction of active compounds is controlled by two-phase boundaries. The first explains the washing step where the compounds are dissolved into the solvent, and the second considers a diffusion step where the solutes from the internal wall cells diffuse into the solvent, this step being slower than the first extraction step due to mass transfer limitations. Different mathematical expressions are used to empirically represent this situation such as the parabolic diffusion model and the double exponential or two-site kinetic model. Mustapa et al. [22] and Natolino and Da Porto [23] used a double exponential model to satisfactorily simulate the extraction curves of phenolic compounds assisted by microwave from the Lindau herb, and assisted by ultrasound from grape marc, respectively.

Other authors have used models with a phenomenological approach. For the extraction of vegetable oils, the broken plus intact cells model has been proposed. This model considers the presence of broken cell walls, usually as a result of a pre-treatment, and intact cells that add an additional mass transfer resistance. In the broken cells, the main mechanism of mass transfer is convection, whereas the intact cell mass transfer is governed by diffusion [24].

To the best of our knowledge, the extraction process of bioactive compounds using power ultrasound from artichokes stems has not been modelled and simulated with a phenomenological model. The main objective of this work was to evaluate the application of acoustic energy as a technology to intensify the bioactive compounds extraction process (phenolic compounds and chlorogenic acid) from artichoke by-products and to propose a mathematical model to simulate the extraction kinetics at different temperatures (25, 40, and 60 °C) and ultrasound power densities (at 200 W/L and 335 W/L), comparing them with conventional mechanical agitation.

## 2. Materials and Methods

### 2.1. Chemicals

The following reagents were used for the different determinations: methanol HPLC grade, ethanol 96% (*v/v*) extra pure, acetonitrile HPLC grade, and Folin-Ciocalteu reagent, which were all from Scharlau (Barcelona, Spain); and orto phosphoric acid 85% (*v/v*) purchased from Panreac (Barcelona, Spain). The standard of chlorogenic acid > 95% was purchased from Sigma Aldrich Company Ltd. (Gillingham, UK).

### 2.2. Sample and Preparation

Artichokes (*Cynara scolymus*) grown in Majorca (Spain), selected with a stem diameter of 1.9 ± 0.2 cm, were purchased from a local supermarket, and stored at 4 °C for a maximum of one week.

To determine the total phenolic compounds content (TPC) and the chlorogenic acid content (CAC) of the artichoke parts, the artichokes were divided into three fractions: stem, bracts, and heart. Each part was frozen at −80 °C in an ultra-freezer (CVF 525/86 Ing. Climas, Spain), subsequently subjected to lyophilization (LyoQuest, Telstar, Spain) at 0.3 mbar and −50 °C, ground (A10, Ika Werke, Germany), sieved (FIT-0200, Filtra, Spain) to a particle size of less than 0.5 mm, vacuum packed (EVT-10, Tecnotrip, Spain) in 20/70 polyamide/polyethylene bags with permeability to O_2_ of 2.58 × 10^−7^ mol/m^2^·s·Pa (Guerrero Coves SL, Valencia, Spain) and stored at 4 °C protected from light until processed (for a maximum of one month).

Methanolic extracts were prepared to determine the TPC and CAC. For this, samples were accurately weighed (~1.0 g), 40 mL of methanol were added, and the mixture was homogenized with the help of an Ultra-Turrax T25 Digital (Ika, Germany) for 60 s, at 13,000 rpm and 4 °C protected from light. The homogenate was refrigerated for 24 h. Subsequently, it was centrifuged at 4000 rpm for 10 min (ALC 4218, Thermo Scientific, Milano, Italy). The supernatant was filtered through Whatman No. 4 paper and stored at 4 °C in the dark until analysis.

The moisture content of the raw and freeze-dried samples was determined according to the official AOAC method 934.06 and the results were expressed as g of water per 1 g of artichoke (on a dry matter basis, dm).

### 2.3. Extraction Experiments

The experimental equipment used to carry out the extraction treatments in liquid medium was a jacketed glass container with a capacity of 600 mL connected to a thermostatic bath (Frigedor, J.P. Selecta, Barcelona, Spain) for temperature control. The selected extraction solvent was water-ethanol (80–20 *v/v*) due to the solubility of phenolic compounds and chlorogenic acid [6,25] and low toxicity and cost, used in a total volume of 200 mL and a solid/solvent ratio of 1:5 (*w/v*, g/mL).

Conventional extraction (A) was conducted with mechanical agitation (100 rpm) by using a conventional stirrer (RZR 2021, Heidolph, Germany) equipped with a four-blade propeller that describes a 50 mm diameter circle. For the extractions with ultrasound assistance (U), the mechanical stirring was replaced by a titanium sonotrode of 100 mm length (Hielscher Ultrasonics, Schwabach, Germany) working in cycles of 0.5 s and immersed 0.5 cm into the extraction solution. The sonotrode was connected to an ultrasonic generator (UP400S, Hielscher Ultrasound Technology, Teltow, Germany). To carry out the extraction experiments, two different sonotrodes of 22 mm (Ua experiments) and 14 mm (Ub experiments) diameter were used, obtaining two different ultrasound power densities which were characterized.

For the extraction experiments, the stem of the artichoke was selected and cut into 0.5 and 1 cm thick slabs. Extractions from 0.5 cm thick artichoke stem slabs were carried out with mechanical agitation and assisted by ultrasound at three different temperatures (25 ± 2 °C, 40 ± 2 °C, and 60 ± 1 °C), because higher temperatures could produce biocompounds’ degradation. The extraction times were 4, 6, 8, 10, 12, 15, 20, 25, 30, and 35 min, since high extraction yields were observed at 35 min. Two additional experiments were carried out with 1 cm thick artichoke stem slabs at 40 °C, the first one with mechanical agitation (100 rpm), and the second one with ultrasound assistance (22 mm diameter sonotrode). In all the experiments, after the corresponding time, 2 mL of extract were taken by syringe, filtered with a 0.45 μm polytetrafluoroethylene (PTFE) filter and stored at 4 °C to subsequently determine both the chlorogenic acid and the total phenolic compounds contents.

### 2.4. Characterization of the Acoustic Field

To measure the acoustic power (W) supplied to the immersion fluid, a calorimetric technique was used. This method consists of measuring the temperature increase during the first 300 s of ultrasound application by using two K-type thermocouple probes connected to a data acquisition equipment HP 34970A Data Logger (COMARK, London, United Kingdom) [26]. The calculation of the ultrasound power was performed using Equation (1), based on the experimentally tripled temperature-time curve.
(1)$$P={\mathrm{MC}}_{p}\frac{\mathrm{dT}}{\mathrm{dt}}$$
where P is the ultrasound power (W), M the mass of solvent (kg) T the temperature (K), t the exposure time (s) and C_p_ the heat capacity of the solvent (J/kg K). The acoustic power densities (W/L) were calculated as the ratio between the ultrasound power (P) and the total extraction volume.

### 2.5. Determination of Total Phenolic Content

The TPC was spectrophotometrically determined, from both the methanolic extracts and those taken at the different points of the extraction process, according to the method of quantification of Folin-Ciocalteu phenols adapted to 96-well microplates described by Eim et al. [27]. The determinations were performed in triplicate and the results were expressed as mg of gallic acid equivalent (GAE) per 1 g of artichoke (on a dry matter basis, dm).

### 2.6. Determination of Chlorogenic Acid Content

The CAC was determined using the method described by Wang et al. [28], using a Thermo-Finnigan Surveyor HPLC analytical system (Thermo Scientific, Paris, France) equipped with a Waters 600E pump, a Waters 2966 photodiode matrix detector and a software-controlled Waters 717 plus autosampler (Empower). For this, 10 μL of extract were injected into a Prodigy 5 µm ODS-3 100 Å (5 μm; 3.2 × 150 mm) (Phenomenex, Torrace, CA, USA). The mobile phases were formed by 0.2% H_3_PO_4_/H_2_O (*v/v*, solvent A) and CH3CN (solvent B). A gradient flow was used starting with 94% A and 6% B and changing to 70% A and 30% B after 20 min. The flow rate was set at 1 mL/min, the temperature at 25 °C and the UV/vis detection wavelength at 330 nm.

The identification and quantification of the main peak were carried out by calibration with external standards (y = 26,837.1x + 13,177.4, R^2^ = 0.9996, LOD = 0.067 mg/L, LOQ = 0.223 mg/L). The experiments were performed in triplicate and the results were expressed in mg chlorogenic acid per 1 g of artichoke (on a dry matter basis, dm).

### 2.7. Mathematical Modelling

The transport of bioactive compounds during the extraction process could take place through different and simultaneous mechanisms that contribute to the total flux. A mathematical model has been proposed to explain the mass transport by combining Fick’s second law with the microscopic mass transfer. A constant and effective diffusion coefficient (D_eff_, m^2^/s) was considered, only temperature-dependent [29], although other mechanisms could coexist. Due to the presence of the external stem skin, it was also assumed that the mass transfer took place mainly through the axial direction; therefore, the solid was considered as an infinite slab. The governing equation obtained is as follows Equation (2):(2)$$\frac{\partial C_{\mathrm{sol}}\left(x,t\right)}{\partial t}=D_{\mathrm{eff}}\frac{\partial^{2}C_{\mathrm{sol}}\left(x,t\right)}{\partial x^{2}}$$
where C_sol_ is the local concentration of the bioactive compound in the solid phase (mg/g dm), x the axis distance (m) and t is the extraction time (s). To solve this partial differential equation, the initial bioactive compounds content (C_o sol_) was considered uniform throughout the slab and both the shape and the size of the slab considered constant during the process. Both solid symmetry and surface convection were the boundary conditions considered.

By using the separation of variables method [30], the following expression (Equation (3)) was obtained for the bioactive compounds content profile in an infinite slab [31], where L (m) is the half-thickness of the slab.
(3)$$\frac{C_{\mathrm{sol}}-C_{e\;\mathrm{sol}}}{C_{o\;\mathrm{sol}}-C_{e\;\mathrm{sol}}}=\overset{\infty }{\underset{\nu =1}{\sum }}\frac{2\;\sin\left(\gamma_{\nu }\right)}{\sin\left(\gamma_{\nu }\right)\;\cos\left(\gamma_{\nu }\right)+\gamma_{\nu }}e^{\left[-\frac{D_{\mathrm{eff}}\;t}{L^{2}}{\;\gamma }_{\nu }{}^{2}\right]\cos\left[\gamma_{\nu }\;\frac{x}{L}\right]}$$
(4)$$\gamma_{\nu }\tan\gamma_{\nu }=\mathrm{Bi}$$
(5)$$\mathrm{Bi}=\frac{h_{m}\;L}{D_{\mathrm{eff}}}$$
where C_e sol_ is the equilibrium bioactive compounds content in the solid phase, and the γ_ν_ coefficients in Equation (3) are the roots of Equation (4). The Biot number (Bi) (Equation (5)) expresses the relative importance of the internal and the external resistances to mass transfer (h_m_, m/s). High values of the Biot number (Bi > 1) mean that the internal transfer is limiting and, therefore, the extraction process is mainly controlled by diffusion. Conversely, low values of the Biot number (Bi < 1) indicate that the external transfer resistance is important. The Biot number calculation allows us to determine if the limiting factor is internal or external. If the internal transfer is limiting, the intensification of surface exchanges will not be affected [32].

The average TPC and CAC in the slab were estimated as the average of the local content (C_sol_) calculated using Equations (3)–(5) in 30 equally distributed positions along L and using enough terms of Equation (3) to achieve a change lower than 1%. The TPC and CAC in the liquid phase at time t were assumed to be the difference between the initial slab contents (C_o sol_) and the estimated average slab contents at that time. The TPC and CAC extraction yields (%) were estimated in percentage as the average contents in the liquid phase divided by the initial contents in the solid phase.

The ‘nlinfit’ function of the optimization toolbox of Matlab R2020b (The MathWorks Inc., Natick, MA, USA), which estimates the coefficients of a nonlinear regression function and the residuals using least squares, was used to identify the model parameters. The mean relative error (MRE, Equation (6)), calculated by comparison of the experimental (C_exp_) and calculated (C_cal_) concentrations, was used as the objective function to minimize. N is the number of experimental data. The MRE has been used in the literature to assess the quality of fit for different mathematical models and it is generally accepted that MRE values below 10% provide a good fit [33].
(6)$$\mathrm{MRE}=\frac{100}{N}\sum_{i=1}^{N}\frac{\left|C_{\exp\;i}-C_{\mathrm{cal}\;i}\right|}{C_{\exp\;i}}$$

### 2.8. Statistical Analysis

For the statistical analysis of the experimental results, R 3.1.0 software [34] together with RStudio IDE [35] were used. Shapiro-Wilk and Levene tests were used to evaluate the normality of data distribution and the homogeneity of variances, respectively. When data were normally distributed and exhibited homogeneity of variances, the parametric ANOVA and Tukey’s tests were used to evaluate the existence and extent of significant differences among samples, respectively. These methods were replaced by robust statistical methods when data were not normally distributed and/or exhibited heterogeneity of variances. In this case, the ‘t1way’ and ‘lincon’ R functions (WRS2 package), a heteroscedastic one-way ANOVA which uses a generalization of Welch’s method, and pairwise comparisons at α = 0.05, respectively [36], were used with trimmed means.

To perform a thorough and objective evaluation of the accuracy of the simulation provided by the proposed mathematical model, the MRE (Equation (6)) and the percentage of explained variance (%var, Equation (7) were determined by a comparison between the experimental results and those provided by the model. S_yx_ and S_y_ are the standard deviations of the estimation and the sample, respectively.
(7)$$\%\mathrm{var}=\left[1-\frac{S_{\mathrm{yx}}}{S_{y}}\right]100$$

## 3. Results and Discussion

### 3.1. Phenolic Compounds of the Artichoke

The TPC and CAC of the three different parts of the artichoke (heart, stem, and bracts) were determined and compared in order to select the best by-product to work with in the subsequent extractions (stem or bracts). The results are shown in Table 1 together with the moisture contents.

According to the FDA, the average moisture content of artichoke is of 5.28 g/g dm [37]. In this study, the moisture content of the three parts exhibited significant (*p* < 0.05) differences. The TPC was lower in the bracts than in both the heart and the stem, with no significant differences (*p* < 0.05) observed between the latter two. The CAC showed a similar trend to that observed in the phenolic compounds, being significantly lower (*p* < 0.05) in the bracts than in the other two parts. Lavecchia et al. [38] analyzed artichoke residues and determined the phenolic compounds in the stem and bracts obtaining values of 51.10 ± 0.74 and 24.58 ± 0.57 mg GAE/g dm, respectively. Song et al. [39] analyzed the chlorogenic acid content of the heart, obtaining a value of 23.81 ± 0.01 mg/g dm.

The heart is the consumed part and therefore is not a by-product. The TPC and CAC of the stem were similar to those in the artichoke heart and significantly (*p* < 0.05) higher than those in the bracts. Therefore, the stem was selected as the raw material for the extraction experiments.

### 3.2. Ultrasonic Power Characterization

The acoustic power density (Pot, W/L) was calculated, as described previously, as the ratio between the ultrasound power applied (P) and the total extraction volume. The measured ultrasound power densities were 200 ± 5 W/L for the 22 mm diameter sonotrode (Ua experiments) and 335 ± 31 W/L for the 14 mm diameter sonotrode (Ub experiments).

### 3.3. TPC and CAC Extraction Yields

The experiments of bioactive compounds extraction from the artichoke stem slabs (0.5 cm thickness) were carried out at different temperatures (25, 40, and 60 °C) with mechanical stirring (A) and with acoustic assistance (Ua at 200 W/L and Ub at 335 W/L) for 35 min. The TPC and CAC were measured in the extracts at different times, and the yields of extraction were estimated as the percentages of the initial contents in the artichoke stem that had moved from the solid matrix to the solvent. These results are shown in Figure 1 as average values and deviations (dots).

A total of three groups of experiments can be observed in these figures depending on the temperature. The extraction yield curves at 25 °C were located at the bottom of each figure, those at 60 °C at the top, and the curves of experiments at 40 °C were located in the middle areas. Thus, the influence of temperature on the extraction yield was remarkable. Moreover, in each of these groups, the extraction yield increased when ultrasound was applied, the increase being higher when the ultrasound power density was higher.

The TPC yield increased, after 35 min of extraction with agitation, from 4–68% when the temperature increased from 25–60 °C. In the case of the ultrasound-assisted extraction, the yield increased from 13–82% in Ua experiments (200 W/L) and from 24–99% in Ub experiments (335 W/L) for the same time and temperature interval. With regard to the CAC extraction yield after 35 min, it increased from 1% at 25 °C to 46% at 60 °C for the experiments with agitation, while the increases in Ua experiments were from 6–60%, and from 8–75% in Ub experiments, for the same temperature increase.

Roselló-Soto et al. [40] observed a 10% increase in phenolic compounds extraction from tiger nut by-products in 50% ethanol at 50 °C with mechanical agitation, in comparison with the same extraction carried out at 25 °C. Irakli et al. [41] carried out the phenolic compounds extraction from olive leaves in an ultrasound bath (frequency 37 kHz) with different solvents and for different times and temperatures (25, 45, and 60 °C). These authors observed that when optimal extraction conditions were applied (50% acetone as solvent and 10 min of extraction time), TPC increased by 9% at 60 °C in comparison with the TPC at 25 °C. Angelov et al. [42] carried out the extraction of phenolic compounds from the stems and the leaves of an artichoke under agitation in thermostatic shaker and observed a 105% increase from the extraction carried out at 20 °C to the extraction carried out at 100 °C.

According to the literature, extracting bioactive compounds has a temperature-labile limit [43], and using temperature above 75 °C may promote significant degradation losses [44]. Moreover, mechanical agitation can be inefficient in extracting the parts of phenolic compounds that can be ester bound or trapped within proteins and polysaccharides, on cell walls [45]. After 35 min of extraction at 25 °C, the TPC and CAC yields increased from 4% and 1%, respectively, in the experiment with agitation, up to 24% and 8%, respectively, when 335 W/L of acoustic energy was applied. The application of ultrasound at 335 W/L caused increases in the TPC and CAC yields, in comparison to the experiments with agitation, of 93% and 67% at 40 °C, and 43% and 62% at 60 °C, respectively. Similar extraction yields of TPC were achieved at 25 °C with the application of power ultrasound (335 W/L) to those obtained at 40 °C by mechanical stirring (100 rpm).

Umaña et al. [33] observed a 193% increase in the ergosterol extraction yield from mushroom stalks in 70% ethanol at 25 °C and 321 W/L of ultrasound power density in comparison to the extraction with mechanical agitation assistance (130 rpm). Arruda et al. [46] carried out the extraction of phenolic compounds and chlorogenic acid from dried araticum peel with 50% ethanol-water (*v/v*), at a maximum temperature of 40.4 °C and applying ultrasound with a sonotrode of 13 mm of diameter, reaching ultrasound power from 160 W to 640 W. These authors reported extraction yields about 41% higher for TPC and 82% higher for CAC when extracting with the highest ultrasound power (640 W) compared with the lowest (160 W).

### 3.4. Mathematical Modelling

The proposed mathematical model parameters were identified for the TPC and CAC extraction curves. In a preliminary identification step, D_eff_, Bi and C_e_ for TPC and CAC extraction kinetics were identified for each tested extraction condition. Once the figures for these parameters were obtained, it was observed that the effective diffusion coefficient was temperature-dependent, according to an Arrhenius type equation (Equation (8)), where D_0_ is the pre-exponential factor (m^2^/s) and E_a_ is the activation energy (J/mol K), R is the gas constant (8.31 J/mol K), and T is the extraction temperature (°C). Moreover, by assigning a zero for the ultrasound power density variable in the experiments carried out with mechanical agitation, the Biot number exhibited a linear dependence with the ultrasound power density (Equation (9)), and the equilibrium content in the liquid phase (extract) was linearly dependent on both the ultrasound power density and the extraction temperature (Equation (10)).
(8)$$D_{\mathrm{eff}}=D_{0}{\;e}^{\left[\frac{E_{a}}{R\left(T+273\right)}\right]}$$
Bi = Bi_0_ + Bi_1_ Pot(9)
C_e_ = C_e0_ + C_e1_ Pot + C_e2_ T(10)

In a second identification step, all the experimental data obtained at the different temperatures, ultrasound power densities, and extraction times were simultaneously used to identify the seven model parameters to simulate the TPC and CAC extraction yields. The identified figures for the parameters of Equations (8)–(10) are shown in Table 2, for both the TPC and CAC extraction kinetics, together with the confidence interval (*p* < 0.05) and standard error for each parameter. As it can be seen, the standard errors were relatively high since a large number of parameters were simultaneously estimated.

According to Equation (8) and figures in Table 2, the effective diffusion coefficient varied from 1.42 × 10^−7^ m^2^/s at 25 °C to 2.43 × 10^−7^ m^2^/s at 60 °C for TPC extraction and from 9.44 × 10^−7^ m^2^/s at 25 °C to 1.79 × 10^−6^ m^2^/s at 60 °C for CAC extraction, thus, increases of 71.6% and 90.1%, respectively.

It can be assumed that the temperature increase favored extraction by improving the solubility of the phenolic compounds. This trend could be observed in the study conducted by Türker and Erdoğdu [47] who concluded that the diffusion coefficient increased by ca 106% in the extraction of pigments from black carrot (*Daucus carota* L.) when the temperature increased from 25 °C (1.8 × 10^−11^ m^2^/s) to 50 °C (3.7 × 10^−11^ m^2^/s).

The estimated activation energy (E_a_) for the TPC and CAC extraction processes was 12.7 kJ/mol and 15.1 kJ/mol, respectively. The parameter E_a_ may depend on different factors related to the extraction process, such as the structures of both the solid matrix and the bioactive compounds, the treatment of the sample before the extraction, the solvent used, among others [48]. According to the literature, when the E_a_ is lower than 20 kJ/mol, the extraction rate is mainly controlled by the diffusion process [49]. Tao et al. [50] estimated 7.0 kJ/mol of activation energy for the extraction of phenolic compounds from grape pomace in an ultrasonic bath system at 25 kHz.

The low figures obtained for the Biot number (from 1.55 × 10^−2^ at 0 W to 1.94 × 10^−2^ at 335 W/L for the TPC extraction, and from 1.31 × 10^−3^ at 0 W to 1.98 × 10^−3^ at 335 W/L for the CAC extraction) are indicative of the relative importance of the external mass transfer resistance. The increase of Biot number was also observed in the extraction of paclitaxel from Taxus Chinensis in an ultrasound bath at 25 °C, by Yoo and Kim [51], from 4.77 at 180 W to 6.50 at 380 W of ultrasound power.

According to Equation (5), the external mass transfer coefficient (h_m_) for the extraction of TPC varied with both the temperature and the ultrasound power density, from 8.79 × 10^−7^ m/s for the extraction by mechanical agitation (0 W/L of ultrasound) at 25 °C to 1.89 × 10^−6^ m/s at 335 W/L and 60 °C, thus a 71.6% increase from 25 °C to 60 °C and a 25.5% increase from 0 to 335 W/L. For the CAC extraction, h_m_ varied from 4.95 × 10^−7^ m/s for the extraction by mechanical agitation (0 W/L of ultrasound) at 25 °C to 1.42 × 10^−6^ m/s at 335 W/L and 60 °C, thus a 90.1% increase from 25–60 °C and a 51.1% increase from 0–335 W/L. In both cases, it can be observed that the effect of the temperature was higher, although the ultrasound assistance also caused significant h_m_ increases.

The equilibrium TPC and CAC in the liquid phase depended on both the ultrasonic power density and the temperature. When the ultrasonic power density increased from 0–335 W/L, C_e_ increased 417%, 73% and 35% for the TPC extraction, and 262%, 33% and 15% for the CAC extraction, at 25, 40, and 60 °C, respectively. Thus, the effect of the ultrasound assistance on C_e_ was higher at lower temperatures. On the other hand, when the temperature increased from 25–60 °C, C_e_ increased by 1103%, 316%, and 214% for the TPC extraction, and 1642%, 641%, and 454% for the CAC extraction, at 0, 200, and 335 W/L of ultrasound power density, respectively.

Using the parameter figures shown in Table 2 together with the model equations (Equations (3)–(5) and (8)–(10)), the extraction curves were simulated for the different extraction conditions and represented in Figure 1 as continuous lines. The accuracy of the simulation obtained with the proposed model and, therefore, its ability to represent the experimental results was also statistically evaluated by using the mean relative error (MRE, Equation (6)) and the percentage of explained variance (%var, Equation (7)). Thus, Table 3 shows the MRE and %var obtained by comparison of the experimental and calculated TPC and CAC in the liquid phase (extracts). It can be seen in Figure 1 that the model provided a satisfactory simulation of the extraction curves. However, as the extraction yield at 25 °C with mechanical agitation was very low even after 35 min (<4% of TPC and <1.1% of CAC), therefore, the accuracy of the simulation was not satisfactory, with MRE>15% and %var < 95%. For the rest of the experiments, the average MRE and %var were 3.7 ± 0.5% and 99.4 ± 0.5% for the simulation of the TPC extraction curves and 6.4 ± 3.0% and 99.3 ± 0.4% for the simulation of the CAC extraction curves.

To further evaluate the robustness of the proposed model, two additional experiments, not used in the model parameters identification, carried out with artichoke slabs of double thickness (1 cm) at 40 °C and agitation (100 rpm) or 200 W/L of ultrasound assistance, were simulated. Figure 2 shows the experimental (dots) and simulated (lines) extraction TPC and CAC yield curves for these experiments. As can be seen in this figure, the simulation provided by the model was satisfactory, even when a different particle size was used. The average MRE and %var were 6.7 ± 2.5% and 99.3 ± 0.5%, respectively, for the TPC extraction, and 5.5 ± 0.2% and 99.3 ± 0.1%, respectively, for the CAC extraction.

Figure 3 shows the simulated vs. experimental TPC and CAC extraction yields for all the experiments carried out with both slab thicknesses, the linear regression of these data (slope and y-intercept) and predicted bounds at 95% of confidence. The coefficients of determination were >0.99, the slopes were not significantly different to 1 (*p* > 0.05) and the y-intercepts were not significantly different to zero (*p* > 0.05), indicating a good agreement between both groups of data, experimental and simulated.

The proposed model was developed by assuming that two resistances in series simultaneously controlled the mass transfer rate during the extraction process, one associated with diffusion and a second one, to convection. To evaluate the necessity of these two resistances, the mathematical model was also solved by considering the external resistance to mass transfer to be negligible. In this case, the model simulation was very unsatisfactory, with MRE > 20% and %var < 80% (results not shown). The artichoke stem is made up of vegetable cells, containing the bioactive compounds mainly in their interior [52]. The major resistance of molecular diffusion in materials of plant origin always comes from the adhering membranes to the intercellular area and cell walls to the liquid phase [53]. Thus, as it is considered in the broken plus intact cells model, two mechanisms of mass transfer can contribute to the extraction kinetics: convection and diffusion. The contribution of each one is different depending on the extraction conditions, agitation or ultrasound assistance, and temperature. It can also change during the extraction time, as the effects of the ultrasound alter the microstructure of the vegetable matrix.

## 4. Conclusions

According to the results obtained in this study, it can be concluded that not only temperature causes a considerable increase in the bioactive compounds extraction yield from artichoke stem, but the ultrasound assistance as well, in comparison with conventional solid-liquid extraction with mechanical agitation assistance. The effects of the ultrasound assistance were greater at lower temperatures, i.e., after 35 min of extraction at 25 °C, the TPC and CAC yields increased by 525% and 700%, respectively, when 335 W/L of ultrasound power density were used, in comparison with the experiments with 100 rpm of mechanical agitation. Taking into account environmental aspects, it could be interesting for the food industry to consider ultrasound as a technology of extraction process intensification since ultrasound application could allow obtaining similar or higher extraction yields of phenolic compounds using lower temperatures and ecofriendly solvents. A phenomenological model has been proposed to simulate the extraction yield curves. This model allowed the quantification of both temperature and ultrasound power density effects in terms of the variation of the model parameters with both variables. The need to consider two series resistance to satisfactorily simulate the extraction yield curves was observed, which could be related to the diffusion of the bioactive compound from inside the vegetable cells toward the intercellular volume and from there, to the liquid phase. The effective diffusion coefficient exhibited temperature dependence according to an Arrhenius equation, whereas the external mass transfer coefficient and the equilibrium extraction yield depended on both the temperature and ultrasound power density. The model allowed the accurate simulation of the extraction yield curves even when different solid particle sizes were used, the average mean relative error of the simulation being of 5.3 ± 2.6%. Finally, the model proposed not only provides insights into the extraction process but is also a useful tool for prediction and optimization.

## Figures and Tables

**Figure 1 foods-10-00931-f001:**
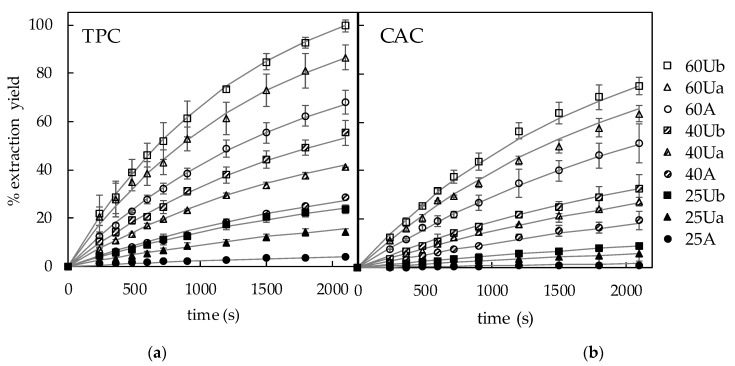
Experimental and simulated extraction yield curves of total phenolic content (TPC) (**a**) and chlorogenic acid content (CAC) (**b**) from artichoke stem slabs (0.5 cm thickness) at different temperatures (25, 40 and 60 °C) with agitation (A, 100 rpm) and acoustic assistance (Ua, 200W/L and Ub, 335 W/L).

**Figure 2 foods-10-00931-f002:**
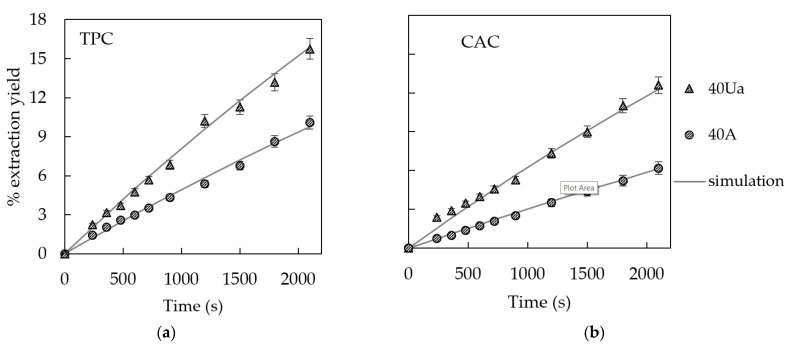
Experimental and simulated extraction yield curves of (**a**) total polyphenol content (TPC) and (**b**) chlorogenic acid content (CAC) from artichoke stem slabs (1 cm thickness) at 40 °C with agitation (A, 100 rpm) and acoustic assistance (Ua, 200 W/L).

**Figure 3 foods-10-00931-f003:**
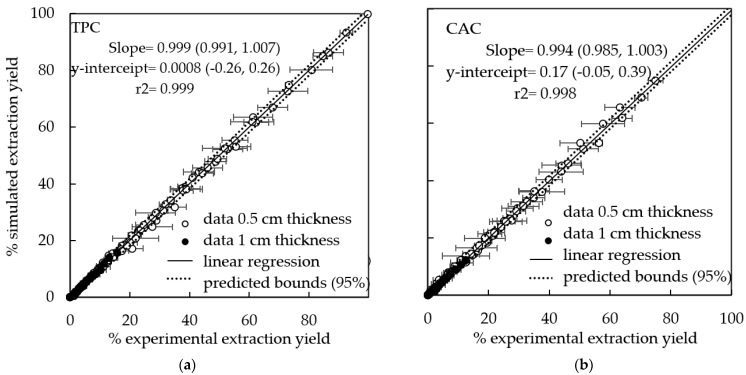
Simulated vs. experimental (**a**) total polyphenol content (TPC) and (**b**) chlorogenic acid content (CAC) extraction yields from artichoke stem, linear regression (slope and y-intercept), and predicted bounds at 95% of confidence. Experiments carried out with slabs of 0.5 cm thickness at different temperatures (25, 40 and 60 °C) with agitation (100 rpm) and acoustic assistance (200 W/L and 335 W/L) and with slabs of 1 cm thickness at 40 °C with agitation (100 rpm) and acoustic assistance at 200 W/L.

**Table 1 foods-10-00931-t001:** Moisture content, total phenolic content (TPC) and chlorogenic acid content (CAC) of the artichoke heart, stem, and bracts.

	Moisture (g Water/g dm)	TPC (mg/g dm)	CAC (mg/g dm)
Heart	4.38 ± 0.20 a	48.9 ± 1.4 a	24.2 ± 1.0 a
Stem	7.00 ± 0.33 c	45.7 ± 2.1 a	21.4 ± 1.1 a
Bracts	5.10 ± 0.29 b	27.4 ± 1.6 b	5.8 ± 1.3 b

Means with different letter for the same column showed significant differences according to the Tukey’s test (*p* < 0.05).

**Table 2 foods-10-00931-t002:** Identified parameters for the proposed model and confidence interval (CI) and standard error (SE) for each parameter. Extraction of total phenolic content (TPC) and chlorogenic acid content (CAC) from artichoke stem at different temperatures under agitation and acoustic assistance.

	TPC			CAC		
	Value	CI	SE	Value	CI	SE
D_o_ (m^2^/s)	2.41 × 10^−5^	[0.93 × 10^−5^, 3.89 × 10^−5^]	6.34 × 10^−6^	4.25 × 10^−4^	[2.73 × 10^−4^, 5.77 × 10^−4^]	2.37 × 10^−1^
E_a_ (kJ/mol K)	12.7	[12.0, 13.5]	4.44 × 10^−5^	15.1	[12.7, 17.5]	1.35 × 10^−4^
Bi_0_	1.55 × 10^−2^	[0.54 × 10^−2^, 2.56 × 10^−2^]	3.36 × 10^−1^	1.31 × 10^−3^	[0.46 × 10^−3^, 2.16 × 10^−3^]	7.33 × 10^−1^
Bi_1_ (L/W)	1.18 × 10^−5^	[0.26 × 10^−5^, 2.15 × 10^−5^]	2.85 × 10^−4^	2.00 × 10^−6^	[0.23 × 10^−6^, 3.77 × 10^−6^]	1.12 × 10^−3^
C_e0_ (mg/g dm)	−24.4	[−25.2, −23.6]	0.39	−12.3	[−12.9, −11.8]	0.27
C_e1_ (mg L/g dm W)	4.41 × 10^−2^	[4.26 × 10^−2^, 4.57 × 10^−2^]	7.8 × 10^−4^	8.98 × 10^−3^	[6.47 × 10^−3^, 11.49 × 10^−3^]	5.3 × 10^−4^
C_e2_ (mg/g dm °C)	1.12	[1.10, 1.13]	7.4 × 10^−3^	0.54	[0.53, 0.55]	5.0 × 10^−3^

**Table 3 foods-10-00931-t003:** Mean relative error (MRE) and percentage of explained variance (%var) estimated by comparison of experimental and simulated extraction kinetics of total phenolic content (TPC) and chlorogenic acid content (CAC) from artichoke stem slabs (0.5 cm thickness) at different temperatures with agitation (A) and ultrasound assistance (Ua, 200W/L and Ub, 335 W/L).

		TPC		CAC	
	T (°C)	MRE (%)	%var	MRE (%)	%var
A	25	>15	<95	>15	<95
	40	6.2	98.5	8.3	98.5
	60	1.9	99.9	1.8	99.9
Ua	25	5.4	98.9	9.7	99.2
	40	1.9	99.8	8.3	99.1
	60	4.5	99.5	3.7	99.3
Ub	25	4.7	99.4	7.8	99.2
	40	3.0	99.6	8.5	99.3
	60	1.9	99.9	3.3	99.6
	Average:	3.7 ± 0.5	99.4 ± 0.5	6.4 ± 3.0	99.3 ± 0.4

## Data Availability

Data is contained within the article.

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
