# Peer review of "Mathematical Modelling of Ultrasound-Assisted Extraction Kinetics of Bioactive Compounds from Artichoke By-Products"

_foods, 2021, doi:10.3390/foods10050931_

Round 1
Reviewer 1 Report
The study investigated the effects of ultrasound-assisted extraction on the total phenolic content (TPC) and chlorogenic acid content (CAC) in the extracts of artichoke stem. Mechanical agitation at 100 rpm was used as the control. Ultrasound assistance were applied at 200 or 335 W/L. The extraction was performed at three temperatures, 25 °C, 40°C, and 60°C. Extracted TPC and CAC were analyzed during the extraction at 4, 6, 8, 10, 12, 15, 20, 25, 30, and 35 min. A mathematical model was proposed to simulate the extraction curve. The study was properly designed and clearly presented. A few suggestions are listed below:
L151: Even though meter (m) is the SI unit. It will be more clear to use 0.5 cm and 1 cm instead of 5·10-3 and 10-2 m.
L353, 355: Indicate the confidence level.
Table 2: The standard errors for multiple parameters were higher than the value. Include discussions about the SEs and what it means for the model.
Author Response
Dear Reviewer,
We are very grateful for all the comments and suggestions relating to our manuscript (foods-1170467). There is no doubt that they have helped to improve the scientific quality of the manuscript.
In general, we are in agreement with the changes suggested. Therefore, we have modified the manuscript accordingly. All comments and corrections are listed in detail below.
- L151: Even though meter (m) is the SI unit. It will be more clear to use 0.5 cm and 1 cm instead of 5·10-3 and 10-2 m.
We agree and have modified the units of the thick slabs in the entire document.
For the extraction experiments, the stem of the artichoke was selected and cut into 0.5 and 1.0 cm thick slabs.
- L353, 355: Indicate the confidence level.
The confidence level has been added in the sentence Line: 360
- Table 2: The standard errors for multiple parameters were higher than the value. Include discussions about the SEs and what it means for the model.
Thank you for your observation. As the number of simultaneously identified parameters was high, the standard errors were also high. However, due to the good simulation of the experimental data and agreement with the physical behavior of the system, we considered that the model is satisfactory. To clarify this, the following sentence has been added. Lines:361-362:
As it can be seen, the standard errors were relatively high since a large number of parameters were simultaneously estimated.

Reviewer 2 Report
The article studies mathematical modelling of ultrasound-assisted extraction kinetics of bioactive compounds from artichoke by-products. The work is interesting, however, there are some points that need to be clarified or supplemented.
- Introduction, line 32: Cynara scolymus L. should be in italics - Cynara scolymus
- Sample and Preparation, line 113: Cynara scolymus should be Cynara scolymus
- Extraction Experiments: How did the authors inspect, that the extraction was exhaustive?
- Determination of Chlorogenic Acid Content: The manuscript doesn't contain validation of chromatographic procedure used in experiment (LOD, LOQ, precision…).
The presented research is extremely simple. Analytics is poor (only tagged: total content of polyphenolic compounds, determination of chlorogenic acid content) . I think that this is not material for an article, and can be publish after deep corrections as short communication.
Author Response
Dear Reviewer,
We are very grateful for all the comments and suggestions relating to our manuscript (foods-1170467). There is no doubt that they have helped to improve the scientific quality of the manuscript.
In general, we are in agreement with the changes suggested. Therefore, we have modified the manuscript accordingly. All comments and corrections are listed in detail below.
This manuscript reports the proposal of a phenomenological mathematical model based on Fick's second law to explain the extraction of biocompounds from artichoke, specifically phenolic compounds and chlorogenic acid, added to the intensification of the process using ultrasound. To test the model, extractions of these biocompounds were carried out using a different thickness of film. Therefore, it is not only based on the extraction of biocompounds, but also focuses on the proposed model.
- Introduction, line 32: Cynara scolymus L. should be in italics - Cynara scolymus
- Sample and Preparation, line 113: Cynara scolymus should be Cynara scolymus
Thank you for your observation. The term Cynara Scolymus has been changed to italics. Lines: 33 and 114.
- Extraction Experiments: How did the authors inspect, that the extraction was exhaustive?
We think that the extraction was not complete. Moreover, as it has been previously demonstrated in the literature, the TPC can vary due to processing probably because of the breakage of more complex molecules and the liberation of simpler phenolic structures, thus, even when the extraction yield could be of 100 %, this does not mean that the remaining TPC is zero [1,2].
Nevertheless, the objective of the study was to obtain the extraction kinetics and to evaluate the influence of ultrasound on these kinetics. To obtain a complete extraction, i.e. to reach a 100% yield, the use of an operation with more than one stage could be advisable.
- Determination of Chlorogenic Acid Content: The manuscript doesn't contain validation of chromatographic procedure used in experiment (LOD, LOQ, precision…).
The LOD and LOQ of the HPLC analysis have been added in the text. As it can be seen, the experimental results obtained in this study were reliable as they were much higher than both LOD and LOQ figures. This information has been added to the manuscript in Lines: 194-195:
The identification and quantification of the main peak were carried out by calibration with external standards (y = 26837.1x + 13177.4, R2 = 0.9996, LOD = 0.067 mg/L, LOQ = 0.223 mg/L).
References
- Raman, V.; Abbas, A. Experimental investigations on ultrasound mediated particle breakage. Ultrason. Sonochem. 2008, 15, 55–64, doi:10.1016/j.ultsonch.2006.11.009.
- Chemat, F.; Zill-e-Huma; Khan, M.K. Applications of ultrasound in food technology: Processing, preservation and extraction. Ultrason. Sonochem. 2011, 18, 813–835, doi:10.1016/J.ULTSONCH.2010.11.023.

Reviewer 3 Report
Development of methods for enhanced extraction of bioactive components from food industry waste and byproducts is one of the key issues of R&D activities. Ultrasound assisted extraction is considered as a promising method for the utilization of bioactive components from artichoke byproduct. Detailed analysis of process parameters on extraction yield and heat- and mass transfer parameters can provide useful and interesting information for the science and practice, as well.
Therefore, the topic of manuscript foods-1170467 can be considered as novel and interesting for the readers.
The manuscript is generally well written with a logic structure. research motivations and the novelties of the study are well defined. Introduction section summarized well the stat-of-art.
Description of analytical methods and modelling and statistical analysis are clear.
Manuscript contains interesting and valuable results that are discussed with relevant references. The developed phenomenological model is utilizable to determine the extraction yield.
Comments, suggestions:
It is not clearly given how was selected the power of ultrasound and extraction time, respectively.
I suggest the authors to improve the quality of figures (Figure 2 and 3) to make the more visible.
Author Response
Dear Reviewer,
We are very grateful for all the comments and suggestions relating to our manuscript (foods-1170467). There is no doubt that they have helped to improve the scientific quality of the manuscript.
In general, we are in agreement with the changes suggested. Therefore, we have modified the manuscript accordingly. All comments and corrections are listed in detail below.
- It is not clearly given how was selected the power of ultrasound and extraction time, respectively.
To establish the extraction conditions, preliminary tests were carried out to determine the best solvent and extraction times. The time of 35 min was considered adequate to study the extraction curves and allowed the comparison among the different operational conditions. Regarding the ultrasound power densities, we selected two different sonotrodes appropriate for the experimental set up which allowed the observation of significant changes in the extraction yields. Then, the ultrasound power densities were experimentally measured for each sonotrode. The following sentences have been modified:
The selected extraction solvent was water-ethanol (80-20 v/v) due to the solubility of phenolic compounds and chlorogenic acid [1,2] and low toxicity and cost, used in a total volume of 200 mL and a solid/solvent ratio of 1:5 (w/v, g/mL). Lines: 139-141
To carry out the extraction experiments, two different sonotrodes of 22 mm (Ua ex-periments) and 14 mm (Ub experiments) diameter were used, obtaining two different ultrasound power densities which were characterized. Lines: 142-152
- I suggest the authors to improve the quality of figures (Figure 2 and 3) to make the more visible.
Thank you for your suggestion. The quality of Figures 2 and 3 has been improved.
References:
- Rabelo, R.S.; Machado, M.T.C.; Martínez, J.; Hubinger, M.D. Ultrasound assisted extraction and nanofiltration of phenolic compounds from artichoke solid wastes. J. Food Eng. 2016, 178, 170–180, doi:10.1016/j.jfoodeng.2016.01.018.
- Zuorro, A.; Maffei, G.; Lavecchia, R. Effect of solvent type and extraction conditions on the recovery of Phenolic compounds from artichoke waste. Chem. Eng. Trans. 2014, 39, 463–468, doi:10.3303/CET1439078.

Reviewer 4 Report
Comments to the Authors/Editor:
The authors of this paper present an interesting study on a Mathematical Modeling of Ultrasound-Assisted Extraction Kinetics of Bioactive Compounds from Artichoke By-Products.
This work could also be improved taking into account the following suggestions:
-Lines 14-16: Please add some numerical results.
-Lines 19: Please add a “.” after the word “observed”.
-The last sentence in the abstract probably has to change with a sentence to highlight the practical or the knowledge gap that your results are covering.
-Lines 97-99: This statement is not exactly correct since with a google search you can see that there is already at least one study for stems and more other studies for artichoke by-products that you not referenced in your study. On the other hand in the discussion section, you use studies for polysaccharides from other plants?
- Please explaining further:
- Why you choose this range of temperature
- Why you choose the time of 35 min.
- Why you choose the range between 200 W/L and 335 W/L
-Lines 106: The Folin chiocalteu reagent is not included in chemicals.
-Line 125: It is correct the range of 0.1g ?? “For this, 1.0 ± 0.1 g of sample”
-Line 166: Please give the explanations of DT and dt after equation 1.
It would be more valuable the estimation of the specific energy (Ev) exposed to your liquid and you can be calculated as follow
Ev=power in W(it depends on the diameter of the sonotrode and the optional supplied power meter) multiply with the time in seconds to have the energy value, followed by dividing the exposed volume in mL. With this will get the Ev in Ws/mL.
-Lines 304-305 and 318-319: The statements are already known.
-in the discussion section you have to compare your results mainly with artichoke by-products extraction studies.Tiger nuts, olive leaves???
- Lines 362-364: The comparison of the extraction of deferent molecules such polyphenols with polysaccharides I think is not correct
-In the conclusion section you have to highlight why your study is important and how the results of this study can have practical application.
Finally, you have to check and correct the similar parts in all the manuscript since the similarity index of Turnitin is 31% .
Author Response
Dear Reviewer,
We are very grateful for all the comments and suggestions relating to our manuscript (foods-1170467). There is no doubt that they have helped to improve the scientific quality of the manuscript.
In general, we are in agreement with the changes suggested. Therefore, we have modified the manuscript accordingly. All comments and corrections are listed in detail below.
- Lines 14-16: Please add some numerical results.
These numbers were not previously included in the abstract because of the 200 words limit. Thus, according to the reviewer’s suggestion, the percent variation of each parameter with the term on which it depends has been added only for the case of the TPC. Moreover, some other words/sentences have been removed or changed from this section.
- Lines 19: Please add a “.” after the word “observed”.
Thank you very much. However, this sentence has been rewritten due to the requirements of another reviewer and accomplish with the 200 words limit for the abstract section.
- The last sentence in the abstract probably has to change with a sentence to highlight the practical or the knowledge gap that your results are covering.
We think that the last sentence in the abstract inherently express our contribution to the extraction mechanisms knowledge. We wanted to show that only one resistance cannot represent the phenomena that take place. The proposed model, taking into account two resistances, was useful to represent the mass transfers in a complex system such as a vegetable structure, probably because there are two serial resistances (movement from inside the cells toward the intercellular volume and from there, to the liquid phase) and also that ultrasound can modify them. However, we have modified the last sentences in order to clarify our conclusion.
- Lines 97-99: This statement is not exactly correct since with a google search you can see that there is already at least one study for stems and more other studies for artichoke by-products that you not referenced in your study. On the other hand in the discussion section, you use studies for polysaccharides from other plants?
Thank you for your observation. It is true that there are some other studies but they use empirical models meanwhile in this sentence, we refer to phenomenological ones. To clarify it, the word “analyze” has been eliminated, so the novelty has been to model and simulate it with a phenomenological model:
To the best of our knowledge, the extraction process of bioactive compounds using power ultrasound, from artichokes stems has not been modelled and simulated with a phenomenological model. Line: 99
- Please explaining further:
Why you choose this range of temperature
Why you choose the time of 35 min.
Why you choose the range between 200 W/L and 335 W/L
To establish the extraction conditions, preliminary tests were carried out to determine the best temperature range, extraction times and ultrasound energy.
Our intention was try to reduce the temperature for economic reasons, but lower than 25ºC was not viable as the extraction yields at this temperature were already low even with ultrasound. Neither temperatures higher than 60ºC seem to be advisable as many bioactive compounds are thermosensitive and degrade.
The time of 35 min was considered adequate to study the extraction curves and allowed the comparison among the different operational conditions.
Regarding the ultrasound power densities, we selected two different sonotrodes appropriate for the experimental set up which allowed the observation of significant changes in the extraction yields. Then, the ultrasound power densities were experimentally measured for each sonotrode.
To carry out the extraction experiments, two different sonotrodes of 22 mm (Ua ex-periments) and 14 mm (Ub experiments) diameter were used, obtaining two different ultrasound power densities which were characterized. Lines: 149-152
Extractions from 0.5 cm thick artichoke stem slabs were carried out with mechanical agitation and assisted by ultrasound at three different temperatures (25 ± 2 °C, 40 ± 2 °C and 60 ± 1 °C), because higher temperatures could produce biocompounds’ degradation. The extraction times were 4, 6, 8, 10, 12, 15, 20, 25, 30 and 35 min since high extraction yields were observed at 35 min. Moreover, this research aimed to reduce the extraction time that would have been required if a conventional extraction was carried out. Lines: 154-158
- Lines 106: The Folin chiocalteu reagent is not included in chemicals.
Thank you for your observation. The Folin-Ciocalteu reagent has been added in chemicals. Lines: 109-110
- Line 125: It is correct the range of 0.1g?
We are very sorry; this was wrongly expressed, and it has been modified as follows:
For this, samples were accurately weighed (∼1.0 g) […]. Lines: 126-127
- Line 166: Please give the explanations of DT and dt after equation 1. It would be more valuable the estimation of the specific energy (Ev) exposed to your liquid and you can be calculated as follow
Ev=power in W(it depends on the diameter of the sonotrode and the optional supplied power meter) multiply with the time in seconds to have the energy value, followed by dividing the exposed volume in mL. With this will get the Ev in Ws/mL.
Thanks for your suggestion. We think that both methods could be used [1,2].
However, we used the calorimetric method because in this method we measure the real energy that ultrasound brings to the set up used in the experimentation, thus, we think it is more reliable and comparable with other studies.
The explanation of T and t has been added.
This method consists of measuring the temperature increase during the first 300 s of ultrasound application […]. Lines: 167-168
where P is the ultrasound power (W), M the mass of solvent (kg) T the temperature (K), t the exposure time (s) and Cp the heat capacity of the solvent (J/kg K). Lines: 172-173
- Lines 304-305 and 318-319: The statements are already known.
Following the recommendation these conclusions have been removed.
- In the discussion section you have to compare your results mainly with artichoke by-products extraction studies.Tiger nuts, olive leaves???
Thank you for your observation. A comparison with an article in which the phenolic compounds are extracted from artichoke by-products has been added:
Angelov et al. [42] carried out the extraction of phenolic compounds from the stems and the leaves of an artichoke under agitation in thermostatic shaker and observed a 105% increase from the extraction carried out at 20 °C to the extraction carried out at 100 °C. Lines: 319-322
- Lines 362-364: The comparison of the extraction of deferent molecules such polyphenols with polysaccharides I think is not correct
The comparison of polyphenol extraction with polysaccharide extraction has been removed.
- In the conclusion section you have to highlight why your study is important and how the results of this study can have practical application.
To highlight the results obtained, so that they can be used in a practical way, the following sentence has been added:
Taking into account environmental aspects, it could be interesting for the food industry to consider ultrasound as a technology of extraction process intensification since ultrasound application could allow obtaining similar or higher extraction yields of phenolic compounds using lower temperatures and ecofriendly solvents. Lines: 473-477
Finally, the model proposed not only provides insights into the extraction process but is also a useful tool for prediction and optimization. Lines: 488-489
- Finally, you have to check and correct the similar parts in all the manuscript since the similarity index of Turnitin is 31%.
Thank you for your observation. We would like to emphasize that this research has not been published in any journal and is not under evaluation in any journal besides Foods Journal.
We also used turnitin before submission to Foods obtaining 27% similarity, and from this, 10% belonged to our own works previously evaluated by turnitin but not published in any paper. In any case, we have now modified the manuscript in order to reduce similarities with other studies.
References:
- Saleh, I.A.; Vinatoru, M.; Mason, T.J.; Abdel-Azim, N.S.; Aboutabl, E.A.; Hammouda, F.M. A possible general mechanism for ultrasound-assisted extraction (UAE) suggested from the results of UAE of chlorogenic acid from Cynara scolymus L. (artichoke) leaves. Ultrason. Sonochem. 2016, 31, 330–336, doi:10.1016/J.ULTSONCH.2016.01.002.
- Minjares-Fuentes, R.; Femenia, A.; Garau, M.C.; Candelas-Cadillo, M.G.; Simal, S.; Rosselló, C. Ultrasound-assisted extraction of hemicelluloses from grape pomace using response surface methodology. Carbohydr. Polym. 2016, 138, 180–191, doi:10.1016/j.carbpol.2015.11.045.

Round 2
Reviewer 2 Report
Authors have done a deep revision of the manuscript. Thus in my opinion this paper should be accepted for publication.
Reviewer 4 Report
Dear authors, you made all the proposed corrections and therefore I think that is ready for publishing. I would like to believe that I helped a little in improving the quality of your article.
Good luck with your research.